# Genomic Instability Signature of Palindromic Non-Coding Somatic Mutations in Bladder Cancer

**DOI:** 10.3390/cancers12102882

**Published:** 2020-10-08

**Authors:** Sophie Vacher, Voreak Suybeng, Elodie Girard, Julien Masliah Planchon, Grégory Thomson, Constance Le Goux, Simon Garinet, Anne Schnitzler, Walid Chemlali, Virginie Firlej, Diane Damotte, Yves Allory, Maud Kamal, Géraldine Pignot, Ivan Bieche

**Affiliations:** 1Unité de Pharmacogénomique, Service de Génétique, Institut Curie, F-75005 Paris, France; sophie.vacher@curie.fr (S.V.); voreak.suybeng@etu.parisdescartes.fr (V.S.); gregory.thomson1@etudiant.univ-reims.fr (G.T.); constance.legoux@gmail.com (C.L.G.); simon.garinet@aphp.fr (S.G.); anne.schnitzler@curie.fr (A.S.); walid.chemlali@curie.fr (W.C.); 2Plateforme de Bioinformatique, Centre de Recherche de l’Institut Curie, F-75005 Paris, France; elodie.girard@curie.fr; 3INSERM U900 (Institut National de la Santé et de la Recherche Médicale), F-75005 Paris, France; 4Unité de Génétique Somatique, Service de Génétique, Institut Curie, F-75005 Paris, France; julien.masliahplanchon@curie.fr; 5TRePCa, Université Paris Est Créteil, F-94010 Créteil, France; virginie.firlej@u-pec.fr; 6Plateforme de ressources biologiques, AP-HP, Hôpital Henri Mondor, F-94010 Créteil, France; 7Service d’Anatomopathologie, Hôpital Cochin, F-75014 Paris, France; diane.damotte@aphp.fr; 8INSERM U1138, Team Cancer, Immune Control, and Escape, Centre de Recherche des Cordeliers, F-75005 Paris, France; 9Département de Pathologie, Institut Curie, F-92210 Saint-Cloud, France; yves.allory@curie.fr; 10Institut Curie, Department of Drug Development and Innovation (D3i), Institut Curie Paris & Saint Cloud, F-75005 Paris, France; maud.kamal@curie.fr; 11Département d’Urologie, Unité de Chirurgie Oncologique, Institut Paoli-Calmettes, F-13009 Marseille, France; pignotg@ipc.unicancer.fr; 12INSERM U1016, Université Paris Descartes, F-75005 Paris, France

**Keywords:** bladder cancer, genomic instability signature, non-coding mutation, clinical biomarkers

## Abstract

**Simple Summary:**

Bladder cancer is the tenth most common cancer worldwide, and its incidence has increased markedly in recent decades. However, current prognostic factors are insufficient to predict outcome at the individual level whereas non-coding somatic alterations remain weakly explored. The goal of this study was to identify clinical biomarkers in non-coding regions for bladder cancer patients. We identified a new type of frequent non-coding somatic genomic instability, specific to bladder tumors. This mutational signature is a promising candidate clinical biomarker for the early detection of relapse and a major low-cost alternative to the TMB to monitor the response to immunotherapy for bladder cancer patients.

**Abstract:**

Numerous pan-genomic studies identified alterations in protein-coding genes and signaling pathways involved in bladder carcinogenesis, while non-coding somatic alterations remain weakly explored. The goal of this study was to identify clinical biomarkers in non-coding regions for bladder cancer patients. We have previously identified in bladder tumors two non-coding mutational hotspots occurring at high frequencies (≥30%). These mutations are located close to the *GPR126* and *PLEKHS1* genes, at the guanine or the cytosine of a TGAACA core motif flanked, on both sides, by a stretch of palindromic sequences. Here, we hypothesize that such a pattern of recurrent non-coding mutations could be a signature of somatic genomic instability specifically involved in bladder cancer. We analyzed 26 additional mutable non-coding sites with the same core motif in a cohort of 103 bladder cancers composed of 44 NMIBC cases and 59 MIBC cases using high-resolution melting (HRM) and Sanger sequencing. Five bladder cancers were additionally analyzed for protein-coding gene mutations using a targeted NGS panel composed of 571 genes. Expression levels of three members of the *APOBEC3* family genes were assessed using real-time quantitative RT-PCR. Non-coding somatic mutations were observed for at least one TGAACA core motif locus in 62.1% (64/103) of bladder tumor samples. These non-coding mutations co-occurred in the bladder tumors but were absent in prostate tumor, HPV-positive Head and Neck Squamous Cell Carcinoma, and high microsatellite instability (MSI-H) colorectal tumor series. This signature of palindromic non-coding somatic mutations, specific to bladder tumors, was not associated with patients’ outcome and was more frequent in females. Interestingly, this signature was associated with high tumor mutational burden (TMB) and high expression levels of *APOBEC3B* and interferon inducible genes. We identified a new type of somatic genomic instability targeting the TGAACA core motif loci flanked by palindromic sequences in bladder cancer. This mutational signature is a promising candidate clinical biomarker for the early detection of relapse and a major low-cost alternative to the TMB to monitor the response to immunotherapy for bladder cancer patients.

## 1. Introduction

Bladder cancer is the tenth most common cancer worldwide, and its incidence has increased markedly in recent decades [1]. About two-thirds of newly diagnosed cases are non-muscle-invasive bladder cancers (NMIBC). These cases have a 60% recurrence rate, and 10% evolve to muscle-invasive tumors. Muscle-invasive bladder cancer (MIBC) represents one-third of cases at diagnosis. Survival greatly differs between early and advanced bladder cancers [2]. Moreover, current prognostic factors, namely tumor node metastasis (TNM) stage and pathological grade, are insufficient to predict outcome at the individual level [3]. New effective molecular markers that may also serve as clinical biomarkers are urgently needed [4].

Pan-genomic studies, using whole-exome sequencing, revealed protein-coding gene alterations that could be used as biomarkers in clinical oncology [5]. Integrated analysis of these various genetic alterations described in TCGA (The Cancer Genome Atlas) revealed three main deregulated signaling pathways in bladder tumors and potential therapeutic targets. Deregulations affecting the cell cycle were found in 93% of cases, those affecting the PI3K/AKT/mTOR pathway were reported in 72% of cases, and those involved in chromatin remodeling impacted 89% of cases [5].

MicroRNAs (miRNAs), a class of small non-coding RNAs, have important roles in the regulation of genes involved in bladder cancer development, progression, and metastasis. Many miRNAs have been studied as potential noninvasive tumor markers, but the diagnostic specificity of miRNAs detection remains to be improved in bladder cancers [6,7].

However, the exome represents only 1–2% of the human genome, and non-coding DNA, representing most of the genome, remains unexplored. Recently, whole genome sequencing analyses have been performed, allowing the study of molecular alterations in non-coding sequences in several cancer types [8,9,10,11,12,13,14,15]. Hotspots and other specific non-coding regions emerged as highly mutated. This is the case for promoter sequences such as *TERT*, long non-coding RNA (lncRNA) such as *NEAT1* or *MALAT1*, and untranslated regions (UTRs) such as *NOTCH1*. The consequences of these non-coding mutations on the expression of corresponding mRNA or protein translation in bladder carcinogenesis (except for *TERT* promoter mutations) remain weakly explored [16].

We have previously identified in bladder tumors two non-coding mutational hotspots occurring at high frequencies (≥30%) within respectively intron 6 of *GPR126* [17] and *PLEKHS1* promoter [18]. The latter two non-coding mutational hotspots had been originally reported at a low frequency (3%) in a cohort of 560 breast cancers [13]. Interestingly, the two non-coding mutational hotspots of intron 6 of *GPR126* and *PLEKHS1* promoter were all located at the guanine or the cytosine of a T**G**AA**C**A core motif that was flanked, on both sides, by a stretch of palindromic sequences.

Here, we hypothesize that such a pattern of recurrent non-coding mutations, co-occurring within a core motif of palindromic sequences, could be a signature of somatic genomic instability specifically involved in bladder cancer.

## 2. Material and Methods

### 2.1. Patients and Samples

We studied a series of 103 bladder cancer patients (composed of 44 NMIBC cases and 59 MIBC cases) who had undergone transurethral bladder resection or a radical cystectomy between January 2002 and January 2007. All patients signed a written informed consent. This study received approval from the local ethics committee (Curie Institute Hospital; Agreement number C75-05-18) and was conducted according to the principles outlined in the Declaration of Helsinki.

Immediately after surgery, tumor samples from each patient were frozen in liquid nitrogen and stored at −80 °C (for DNA and RNA extraction). Tumors were re-staged according to the 2017 TNM classification of bladder tumors and were graded according to the World Health Organization (WHO) 2016 tumor-grading scheme [19]. Standard follow-up visits were performed according to current guidelines. Data were obtained from the patients’ medical records. Complete clinical, histological, and survival information were available for this series.

The cohort consisted of 17 women and 86 men, with a median age of 67.6 years (range 40–91). Pathologic staging identified NMIBC in 44 patients (20 low-grade Ta, 10 high-grade Ta, 14 high-grade T1) and high-grade MIBC in 59 patients. For NMIBC, the mean follow-up was 31 months (range 1–158 months). Among the 44 cases of NMIBC, 22 (50%) had one or more recurrences of NMIBC during the follow-up. Progression to a muscle-invasive tumor was observed in 8 patients (18.2%). For MIBC, the mean follow-up was 29 months (range 1–152 months). During the follow-up period, 31 patients (52.5%) with MIBC died of bladder cancer, and 3 (5.1%) died from unrelated causes. Clinical, histological, biological (including *FGFR3, PIK3CA* and *TERT* mutational status), and survival characteristics of the series of NMIBC and MIBC are presented in Table 1 and Table 2, respectively.

### 2.2. RNA and DNA Extractions

Total RNA was extracted from bladder specimens by using RNAble® (Eurobio, Les Ulis, France) according to the manufacturer’s instructions. The quality of the RNA samples was verified by electrophoresis through agarose gel, staining with SYBR® Safe (Thermo Fisher Scientific, Waltham, MA, USA), and visualization of the 18S and 28S RNA bands under blue light.

Total genomic DNA was extracted with QIAamp DNA Mini kit (Qiagen, Hilden, Germany) following supplier’s recommendations.

Nucleic acids were quantified using a Nanodrop spectrophotometer ND-1000 (ThermoScientific, Wilmington, DE, USA), and purity was assessed with 230/280 ratios for proteins contaminations and 230/260 ratios for phenol contaminations.

### 2.3. In Silico Identification of New TGAACA Core Motif of Palindromic Sequences in the Human Genome 

Nucleotides sequences were extracted from the hg19 reference genome in a window of +/− 30 base pairs (bp) using the R package bedr (https://cran.r-project.org/web/packages/bedr/index.html) around each called single nucleotide variation (SNV). Then, the alteration was included into the sequence. Patterns between 5 and 8 bp presenting both the variant and a palindromic sequence longer than 7 bp on each side were searched inside those windows using R package BioStrings (https://bioconductor.org/packages/release/bioc/html/Biostrings.html). Homopolymers were further discarded.

### 2.4. DNA Non-Coding Mutations Analysis

The assessment of all non-coding mutational hotspots was performed using high-resolution melting (HRM). Samples with an altered HRM profile were validated by Sanger sequencing, which allowed characterizing the mutations. The nucleotide sequences of the primers for the 29 loci tested in this study are listed in Appendix A.

### 2.5. Targeted Next Generation Sequencing (NGS) 

Five tumors were analyzed for protein-coding gene mutations by targeted next-generation sequencing (NGS) that has been recently developed in the genetics department of the Curie Institute. The in-house NGS panel includes 571 genes of interest in oncology for diagnosis, prognosis, and theranostics. Library preparation was performed using the Agilent Sureselect XT HS kit, and sequencing was completed on an Illumina NovaSeq 6000 sequencer. All variants, called using Varscan2 (v2.4.3-0), that passed the following thresholds were validated: allelic ratio above 5% and population frequency lower than 0.1% in 1000 g, ESP or gnomAD.

This large targeted NGS panel also allowed molecular analysis of tumors for CNV (copy number variation) and TMB (tumor mutational burden) status.

### 2.6. Assessment of the Microsatellite Instability (MSI) Status

Microsatellite instability (MSI) testing was performed using the MSI Analysis System (Promega, France), which is composed of 5-mononucleotide repeats (BAT-25, BAT-26, NR-21, NR-24, and MONO-27) to determine MSI status, following the manufacturer’s instructions.

### 2.7. Real-Time Quantitative RT-PCR

The theoretical basis, PCR consumables, and PCR-reaction conditions have been previously described in detail [20]. The precise amount of total RNA added to each reaction mix (based on optical density) and its quality (i.e., lack of extensive degradation) are both difficult to assess. Therefore, we also quantified transcripts of the *TBP* gene (Genbank accession NM_003194) encoding the TATA box-binding protein (a component of the DNA-binding protein complex TFIID) as an endogenous RNA control, and we normalized each sample on the basis of its *TBP* content [21]. Results, expressed as N-fold differences in *APOBEC3* gene expression relative to the *TBP* gene and termed “N*_APOBEC3_*”, were determined as N*_APOBEC3_* = 2^ΔCtsample^, where the ΔCt value of the sample was determined by subtracting the average Ct value of the *APOBEC3* gene from the average Ct value of the *TBP* gene. The N*_APOBEC3_* values of the samples were subsequently normalized such that the median of the N*_APOBEC3_* values for the twenty normal bladder tissues was 1. The primers for *TBP* and the 3 *APOBEC3* family genes were chosen with the assistance of the Oligo 6.0 program (National Biosciences, Plymouth, MN). We scanned the dbEST and nr databases to confirm the total gene specificity of the nucleotide sequences chosen for the primers and the absence of single nucleotide polymorphisms. The primer pairs for each *APOBEC3* gene were selected to be unique with respect to the other *APOBEC3* genes. The nucleotide sequences of the oligonucleotide hybridization primers are shown in Appendix A. To avoid the amplification of contaminating genomic DNA, one of the two primers was placed at the junction between two exons or on two different exons. Agarose gel electrophoresis was used to verify the specificity of PCR amplicons.

### 2.8. Statistical Analysis

Relationships between mutation profiles and clinical histological and biological parameters (including mRNA levels of *APOBEC3* genes and immune genes) were tested using the non-parametric tests, namely the chi-square test, chi-square test with Yates correction and the Fisher test (relation between two qualitative parameters), and the Mann–Whitney test (relation between one qualitative parameter and one quantitative parameter).

For MIBC, overall survival (OS) was calculated from the date of surgery until death or the last follow-up. Recurrence-free survival (RFS) was defined as the time elapsed from the date of surgery until the first local relapse or first metastasis. For NMIBC, progression-free survival (PFS) was defined as the time elapsed from the date of surgery until progression to muscle-invasive disease. Patients were censored if they had not experienced the end-point of interest at the time of the last follow-up. Survival curves were derived from Kaplan–Meier estimates. The log-rank test was used to compare survival distributions between subgroups. Differences were judged significant at a confidence level of >95% (*p* < 0.05).

## 3. Results

### 3.1. Non-Coding Mutations within a TGAACA Core Motif of 10 Palindromic Sequences

We previously identified, in our global cohort of 103 bladder cancer samples, high frequencies of two non-coding mutational hotspots: intron 6 of *GPR126* in 45.6% [17] and *PLEKHS1* promoter in 29.1% [18]. These alterations had originally been reported at a low frequency (3%) in a cohort of 560 breast cancers by Nik-Zainal et al. [13]. In the present study, we analyzed eight additional mutable non-coding sites (TGAACA core motif of palindromic sequences) identified by Nik-Zainal et al. (from Appendix A). In our global cohort of bladder cancer samples, somatic mutations were observed at the same two positions of the core motif (TGAACA) with a range of frequencies of 1% (1/103) for Chr3:82 locus (genomic position: chr3:82807069) to 11.7% (12/103) for Intron ADM locus (genomic positions: chr11:10331381 and chr11:10331384) (Figure 1 and Figure 2, Appendix A). Almost all mutations were G > A and C > T mutations (Figure 2). All the identified variants were extremely rare (<0.0001) or absent in the gnomAD database gathering genomic data of 15,708 whole genome sequences from unrelated individuals (gnomAD v2.1.1; http://gnomad.broadinstitute.org/), confirming the somatic feature of these mutations (Appendix A).

In order to assess the specificity of the non-coding mutations identified in our 103 bladder cancer series, we checked whether the five most frequently mutated non-coding sites (i.e., GPR126, PLEKHS1, Intron ADM, Chr7:11, and Chr15:96) were present in different series of 51 prostate tumors, 10 colorectal tumors with high microsatellite instability (MSI-H), and 10 human papilloma virus (HPV)-positive Head and Neck Squamous Cell Carcinoma (HNSCC). Interestingly, no mutation was observed in these three tumor types for these five mutable non-coding sites.

To test a possible common molecular mechanism of tumor genomic instability that could explain these high frequencies of palindromic non-coding somatic mutations at the core motif (TGAACA) specifically observed in bladder cancer, we sought positive associations between the five most frequently mutated non-coding sites (i.e., GPR126, PLEKHS1, Intron ADM, Chr7:11, and Chr15:96). Several high significant positive associations were found, in particular between Intron ADM mutations and PLEKHS1 mutations (*p* = 0.0007), and between Intron ADM mutations and Chr15:96 mutations (*p* = 0.0004) (Table 3). 

### 3.2. Additional Non-Coding Mutations within a TGAACA Core Motif of Palindromic Sequences Identified by in Silico Analysis of the Human Genome

We next searched, by in silico analysis of the human genome (as detailed in Material and Methods section), for all the other loci that contained the same TGAACA core motif of palindromic sequences of 9, 10, or 11 base pairs (bp) in length, in order to screen for such additional non-coding somatic mutations in bladder cancer. Overall, beside the 10 loci previously described [13], we identified in the human genome 83 additional loci that displayed a TGAACA core motif of palindromic sequences of 9, 10, or 11 pb (Appendix A). All these 83 loci were in the non-coding genome.

Among the 83 loci, we selected 18 of them that were outside repetitive sequences (mostly LINE and Alu repeats) and which could be easily PCR-amplified for sequencing (Appendix A). In a first step, we assessed these 18 loci in a screening set of 20 bladder cancer series. Somatic mutations were observed (at the same two positions of the core motif TGAACA) for eight out of the 18 tested loci in at least one of the 20 tumor samples. For these eight mutated loci, we analyzed, in a second step, the remaining 83 bladder tumor samples. Somatic mutations were observed at the same two positions of the core motif (TGAACA) with frequencies ranging from 1% (1/103) for *CLVS2* locus (genomic position: chr6:123442661) to 6.8% (7/103) for GABRG3 locus (genomic positions: chr15:27617168 and 15:27617171) (Figure 1 and Figure 2, Appendix A). All these variants were also very rare or absent in the gnomAD database (Appendix A).

Thus, overall, somatic mutations were observed for at least one out of the 18 TGAACA core motif loci (10 previously published and eight newly identified) in 62.1% (64/103) of our bladder tumor samples. Two tumors (i.e., T254 and T331) showed up to eight mutated loci among the 18 analyzed loci (Figure 1).

### 3.3. Non-Coding Mutations within an AGATCA Core Motif of Palindromic Sequences in an Intron of RAD51B

Beyond the TGAACA core motif of palindromic sequences, we also checked whether non-coding mutations could also occur within another 6 bp core motif flanked by palindromic sequences in bladder cancer. Thus, we focused on an AGATCA core motif of a 9 bp palindromic sequence in intron 10 of RAD51B, which was originally reported to be specifically mutated in breast cancer [12]. Somatic mutations were observed in 4.9% (5/103) of bladder tumor samples (Figure 1 and Figure 2, Appendix A). The nature and the positions of the two mutated bases were the same in the two core motifs (AGATCA and TGAACA, respectively), and they were all G > A or C > T mutations. Of note, the five mutated bladder tumors in the AGATCA core motif of RAD51B were also mutated for at least one of the 18 TGAACA core motif loci (Figure 1). This result suggests that the putative molecular mechanism of genomic instability in bladder cancer might not be restricted to the AGATCA core motif of palindromic sequences in the human genome.

### 3.4. Relationship between APOBEC3 RNA Level and TGAACA Core Motif Mutations

Globally, the existence of co-mutations on these different loci (located on different chromosomes) suggests an unknown molecular mechanism of genomic instability (different from the microsatellite instability) in bladder cancer. All these mutated tumors mainly showed base substitutions typical of the mutational trinucleotide signature 2 (SBS2) attributed to APOBEC (Apolipoprotein B mRNA Editing Catalytic Polypeptide-like) activity characterized by Alexandrov et al. [22]. APOBEC DNA-editing proteins target the TCN sequence motif, and in particular, the TCA sequence (or the reverse complement TGA) with predominantly C>T substitutions (or the reverse complement G > A) as observed in our bladder tumor series (Figure 2). The other mutations observed in our bladder tumor series were also mostly C > T substitutions (or the reverse complement, G > A) but within the ACA sequences (or the reverse complement, TGT).

Consequently, we tested the possible link between our TGAACA core motif mutational signature and the expression of three members of the *APOBEC3* family: *APOBEC3A, APOBEC3B,* and *APOBEC3H*. Patients were subdivided into three groups: group 1 with tumors showing an absence of TGAACA mutations (*n* = 39), group 2 with one or two TGAACA mutations (*n* = 46), and group 3 with three or more TGAACA mutations (*n* = 18). High rates of TGAACA mutations were significantly associated with high expression levels of *APOBEC3B* (*p* = 0.044), but not with those of the two other *APOBEC3* members (Table 4). *APOBEC3B* expression levels were 2-fold higher in tumors showing high rates of TGAACA mutations, as compared to other tumors.

### 3.5. Association between TGAACA Core Motif Mutations and Clinico-Biological Parameters

The distribution of these groups of TGAACA mutations, according to their clinical, histological, and biological (including *FGFR3, PIK3CA,* and *TERT* mutational status) characteristics, is presented in Table 5. A significant association was observed between a high level of TGAACA mutations and female patients (*p* = 0.0017). Surprisingly, the NMIBC patients and the *FGFR3*-mutated patients were more frequent in group 2, showing a moderate level of TGAACA mutations (one or two TGAACA mutations) (*p* = 0.039 and *p* = 0.015, respectively). Of note, the association between TGAACA mutations and *TERT* promoter mutations that are located in a non-coding region, but without palindromic sequences was not significant (only a trend toward a positive association; *p* = 0.08).

The outcomes of patients in the three groups of TGAACA mutations did not differ in terms of RFS in the global population and PFS in the NMIBC subgroup, as well as RFS and OS in the MIBC subgroup (Appendix A).

### 3.6. Protein-Coding Gene Mutations in Bladder Tumors Showing High Levels of Non-Coding TGAACA Mutations

Four bladder tumors (T331, T206, T254, and T238 from Figure 1) showing high levels of non-coding TGAACA mutations (and one control bladder tumor—T272—with the absence of non-coding TGAACA mutations) were analyzed for protein-coding gene mutations using an in-house targeted NGS panel. The number of non-synonymous mutations in these four highly non-coding TGAACA mutated tumors was ranging from 52 to 93 variants per tumor (Appendix A). The T331 tumor with 78 variants showed high microsatellite instability (MSI-H). The four tumors showed high TMB, as compared to the T272 control bladder tumor.

Pathogenic variants in the 571 cancer genes covered by the in-house targeted NGS for these five tumors are also described in Appendix A. Interestingly, the MSI-H tumor (T331) showed two somatic non-sense mutations for the *MLH1* gene.

### 3.7. Association between TGAACA Core Motif Mutational Signature and Expression Levels of Immune-Related Genes

Tumor mutational burden (TMB) has been associated with expression levels of immune-related genes and response to immunotherapy [23]. Consequently, we tested the possible link between our TGAACA core motif mutational signature and the expressions of 57 immune-related genes that were previously analyzed in our bladder tumor series [24]. High rates of TGAACA mutations were significantly associated with high expression levels of the subgroup of the interferon inducible genes (Appendix A).

## 4. Discussion

Recently, whole-genome sequencing studies have focused on tumor non-coding genomes in different cancer types [8,9,10,11,12,13,14,15]. Although these studies remain rare due to the economic cost and complexity of whole-genome analyses, they identified somatic alteration hotspots in various non-coding regions, including promoters, long ncRNA, UTR, intronic, and intergenic sequences. These recurrent alterations can now be investigated by targeted techniques in various cancers, especially to establish their clinical utility.

In the present study, we identified a new type of somatic genomic instability (independent of the classical microsatellite instability), targeting the TGAACA core motif loci flanked by palindromic sequences, in bladder cancer. This signature of palindromic non-coding somatic mutations was observed in 62.1% (64/103) of our bladder tumors cohort. This mutational TGAACA core motif signature were extremely rare or absent from the genome Aggregation Database (gnomAD; http://gnomad.broadinstitute.org/), confirming the somatic feature of this mutational signature.

This mutational TGAACA core motif signature seems associated with the mutational trinucleotide signature 2 attributed to APOBEC DNA-editing proteins activity, with many base substitutions sharing a characteristic sequence context (C > T substitution at TCN sequence motif, and in particular the TCA sequence or this reverse complement TGA). The APOBEC (Apolipoprotein B mRNA Editing Catalytic Polypeptide-like) family of proteins has diverse and important functions in human health and disease. These proteins have an intrinsic ability to bind to both RNA and single-stranded (ss)DNA. The common core of APOBEC structures is the cytidine deaminase domain, which converts cytidine to uracil. The APOBEC family of cytidine deaminases represents a major enzymatic source of mutations in cancer [22]. The mutational TGAACA core motif signature was rarely observed in breast cancer [13] and absent in HPV-positive HNSCC (the present study), which are two cancer types well-known to be associated with mutational trinucleotide signature 2 [22,25]. This atypical mutational TGAACA core motif signature that seems restricted to bladder cancer is challenging and requires further investigation.

Our results suggest that a high level of mutations occurs in non-coding regions of bladder tumor genomes, but the part played by these hotspots of non-coding mutations in bladder carcinogenesis remains to be fully understood. Except for *TERT* promoter mutations, most of the known non-coding mutations identified as recurrent in cancer did not associate significantly with mRNA expression level changes of target genes located nearby on the genome [15]. This suggests that the effects of these non-coding mutations are through mechanisms outside of gene transcriptional regulation. Many of known non-coding mutations are not yet widely appreciated as cancer driver mutations, motivating further studies on the mechanistic basis of this mutation type in cancers. Conversely, Buisson et al. [14] suggested that the mutations in DNA loci that can form hairpins (such as mutations identified in the present study) could be passengers and driven by *APOBEC3A* [14]. This is in accordance with our results, where high rates of TGAACA mutations were significantly associated with high expression levels of *APOBEC3B* in our bladder tumor samples, but not with those of the *APOBEC3A.* The interrogation and interpretation of non-coding mutations in cancers will become more accurate with the increasing availability of whole-genome sequencing data.

An interesting application of such a frequent somatic non-coding mutations (≈60%) is that they could be considered as a clinical biomarker in bladder cancer. This marker could be easily tested in circulating tumor DNA (ctDNA), which emerged as a promising biomarker in oncology [26,27]. Our mutational TGAACA core motif signature (unlike the *FGFR3* mutations) occurs in both NMIBC and MIBC, rendering it relevant as either a diagnostic or minimal residual disease markers. Non-coding mutations detected in ctDNA or in urine would provide an additional argument for the diagnosis of malignancy or early detection of relapse.

This mutational signature does not seem to be of prognostic interest, since it was not statistically associated with outcome in our bladder tumor series. Finally, this mutational signature could be a major economic low-cost alternative to biomarkers that are currently used in the clinic to monitor response to immunotherapy in bladder cancer patients such as the tumor mutational burden (high TMB, ≥10 mutations per megabase) [28]. To test this hypothesis, it would be necessary to conduct a prospective randomized clinical study to show that this mutational signature does influence outcome only in patients who received immunotherapy as compared to untreated patients. Given the potential toxicity of immunotherapy and the highly variable response to immune checkpoint inhibitors, as well as the significant economic cost of these agents, there is an urgent need for new biomarkers that could predict the response to immunotherapy. Our data are consistent with the hypothesis that a high level of non-coding mutations from our mutational signature is associated with high TMB and high expression levels of *APOBEC3B* and some immune-related genes—in particular, the subgroup of interferon inducible genes. In this regard, *APOBEC3B* (but not *APOBEC3A*) expression has been recently positively associated with interferon inducible genes expression in lung adenocarcinoma [29]. These authors have shown that interferon pathways were enriched in tumors with high APOBEC mutagenesis and that the treatment with IFN-γ led to a significant increase in the expression of *APOBEC3B* in lung epithelial cell lines. These results suggest that IFN-signaling *via* the tumor microenvironment is a potential mechanism of mutational heterogeneity in lung tumors with increased *APOBEC3B* transcripts expression.

## 5. Conclusions

In conclusion, we identified a new type of genomic instability targeting the TGAACA core motif loci flanked by palindromic sequences that is specific to bladder cancer. Further studies are necessary to identify the cause and mechanisms of this genomic instability. This mutational signature of palindromic non-coding somatic mutations is a promising clinical biomarker for the early detection of relapse and a major low-cost alternative to the tumor mutational burden (TMB) to monitor the response to immunotherapy in bladder cancer patients by detecting it in circulating tumor DNA in blood or urine.

## Figures and Tables

**Figure 1 cancers-12-02882-f001:**
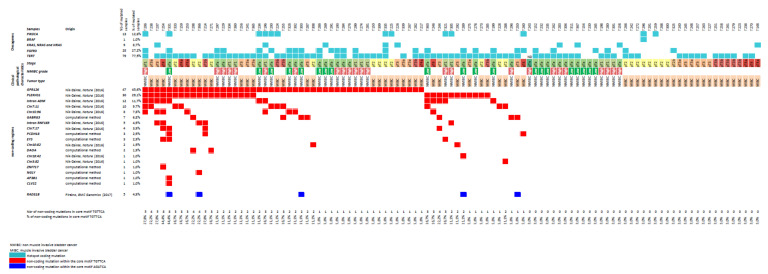
Non-coding mutation landscape, and clinical and biological characteristics of the bladder cancer series.

**Figure 2 cancers-12-02882-f002:**
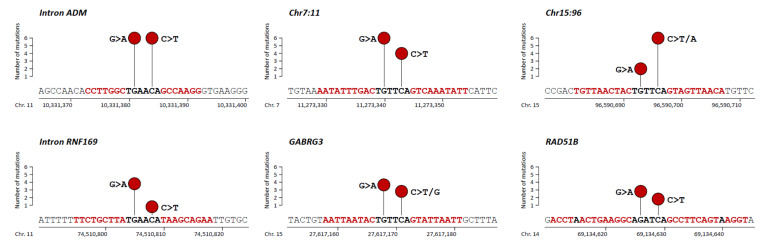
Needle-plots of recurrent mutations along the sequences of selected non-coding and palindromic regions. Palindromic sequences are highlighted in red.

**Table 1 cancers-12-02882-t001:** Clinical, biological, pathological, and survival characteristics of the 44 non-muscle-invasive bladder cancers (NMIBC).

Parameters	Whole Population	No Recurrence	Recurrence		Muscle-Invasive Progression	
	Number of Patients (%)	Number (%)	Number (%)	*p*-Value *	Number (%)	*p*-Value **
Total population	44 (100.0)	14 (31.8)	22 (50.0)		8 (18.2)	
**Age (years)**						
≥60	34 (77.3)	10 (29.4)	16 (47.1)	0.77	8 (23.5)	0.17
<60	10 (22.7)	4 (40.0)	6 (60.0)		0 (0.0)	
**Sex**						
Male	41 (93.2)	13 (31.7)	20 (48.8)	>0.99	8 (19.5)	>0.99
Female	3 (6.8)	1 (33.3)	2 (66.7)		0 (0.0)	
**Smoking status**						
Non-smoker	19 (43.2)	4 (21.1)	12 (63.2)	0.13	3 (15.8)	0.97
Smoker	25 (56.8)	10 (40.0)	10 (40.0)		5 (20.0)	
**History of NMIBC**						
No	24 (54.5)	11 (45.8)	10 (41.7)	**0.049**	3 (12.5)	0.50
Yes	20 (45.5)	3 (15.0)	12 (60.0)		5 (25.0)	
**Associated pTis**						
No	42 (95.4)	14 (33.3)	22 (52.4)	NA	6 (14.3)	**0.030**
Yes	2 (4.6)	0 (0.0)	0 (0.0)		2 (100)	
**Grade**						
Low grade	20 (45.5)	9 (45.0)	10 (50.0)	0.27	1 (5.0)	0.094
High grade	24 (54.5)	5 (20.8)	12 (50.0)		7 (29.2)	
**Tumor stage**						
Ta	30 (68.2)	10 (33.3)	18 (60.0)	0.75	2 (6.7)	**0.008**
T1	14 (31.8)	4 (28.6)	4 (28.6)		6 (42.9)	
***FGFR3* status**						
Mutated	22 (50.0)	5 (22.7)	16 (72.7)	**0.028**	1 (4.5)	0.051
Not mutated	22 (50.0)	9 (40.9)	6 (27.3)		7 (31.8)	
***PIK3CA* status**						
Mutated	7 (15.9)	4 (57.1)	2 (28.6)	0.18	1 (14.3)	>0.99
Not mutated	37 (84.1)	10 (27.0)	20 (54.1)		7 (18.9)	
***TERT* status ^a^**						
Mutated	35 (81.4)	10 (28.6)	18 (51.4)	0.75	7 (20.0)	0.32
Not mutated	8 (18.6)	4 (50.0)	4 (50.0)		0 (0.0)	

* Chi^2^ test, Chi^2^ test with Yates’ correction or Fisher test if appropriate (recurrence vs. no recurrence); ** Chi^2^ test, Chi^2^ test with Yates’ correction or Fisher test if appropriate (muscle-invasive progression vs. others); In bold: *****
*p*-value < 0.05; ^a^ Data available for 43 patients; NA: not applicable.

**Table 2 cancers-12-02882-t002:** Clinical, biological, pathological and survival characteristics of the 59 muscle-invasive bladder cancer (MIBC).

Parameters	Whole Population	Recurrence-Free Survival		Overall Survival	
	Number of Patients (%)	Number of Events (%) ^a^	*p*-Value *	Number of Events (%) ^b^	*p*-value *
Total population	59 (100.0)	39 (66.1)		34 (57.6)	
**Age (years)**					
≥60	43 (72.8)	33 (76.7)	**0.0046**	30 (69.8)	**0.002**
<60	16 (27.2)	6 (37.5)		4 (25.0)	
**Sex**					
Male	45 (76.3)	27 (60.0)	0.15	26 (57.8)	0.97
Female	14 (23.7)	12 (85.7)		8 (57.1)	
**Smoking status ^c^**					
Non-smoker	9 (17.0)	7 (77.8)	0.50	7 (77.8)	0.25
Smoker	44 (83.0)	26 (59.1)		22 (50.0)	
**History of NMIBC**					
No	35 (59.3)	21 (60.0)	0.23	20 (57.1)	0.93
Yes	24 (40.7)	18 (75.0)		14 (58.3)	
**Associated pTis**					
No	52 (88.1)	36 (69.2)	0.21	32 (61.5)	0.12
Yes	7 (11.9)	3 (42.9)		2 (28.6)	
**Tumor stage**					
T2	21 (35.6)	13 (61.9)	0.61	8 (38.1)	**0.024**
≥T3	38 (64.4)	26 (68.4)		26 (68.4)	
**Lymph node status ^d^**					
N−	37 (64.9)	20 (54.1)	**0.019**	17 (45.9)	**0.004**
N+	20 (35.1)	17 (85.0)		17 (85.0)	
***FGFR3* status**					
Mutated	6 (10.2)	4 (66.7)	>0.99	4 (66.7)	>0.99
Not mutated	53 (89.9)	35 (66.0)		30 (56.6)	
***PIK3CA* status**					
Mutated	6 (10.2)	2 (33.3)	0.17	2 (33.3)	0.39
Not mutated	53 (89.9)	37 (69.8)		32 (60.4)	
***TERT* status**					
Mutated	44 (74.6)	31 (70.5)	0.23	26 (59.1)	0.70
Not mutated	15 (25.4)	8 (53.3)		8 (53.3)	

* Chi^2^ test, Chi^2^ test with Yates’ correction or Fisher test if appropriate; In bold: *****
*p*-value < 0.05; ^a^ first recurrence (local or metastatic); ^b^ death; ^c^ data available for 53 patients; ^d^ data available for 57 patients.

**Table 3 cancers-12-02882-t003:** Co-occuring mutations between the 5 most frequently mutated non-coding sites.

Non-Coding Sites	Status	GPR126				PLEKHS1				Intron ADM				Chr7:11			
		WT	M	Total	*p*-Value *	WT	M	Total	*p*-Value *	WT	M	Total	*p*-Value *	WT	M	Total	*p*-Value *
**PLEKHS1**	WT	45	28	73	**0.021**												
	M	11	19	30													
	Total	56	47	103													
**Intron ADM**	WT	51	40	91	0.35 (NS)	70	21	91	**0.0007**								
	M	5	7	12		3	9	12									
	Total	56	47	103		73	30	103									
**Chr7:11**	WT	53	40	93	0.20 (NS)	68	25	93	0.15 (NS)	84	9	93	0.091 (NS)				
	M	3	7	10		5	5	10		7	3	10					
	Total	56	47	103		73	30	103		91	12	103					
**Chr15:96**	WT	55	40	95	**0.035**	70	25	95	**0.045**	88	7	95	**0.0004**	86	9	95	0.57 (NS)
	M	1	7	8		3	5	8		3	5	8		7	1	8	
	Total	56	47	103		73	30	103		91	12	103		93	10	103	

In bold: * *p*-value < 0.05.

**Table 4 cancers-12-02882-t004:** Associations between TGAACA core motif mutations and mRNA expression levels of *APOBEC3* genes.

Gene	Normal Bladder Tissues (*n* = 19)			Group 1 - Bladder Tumors No non-Coding Mutations (*n* = 39)			Group 2 - Bladder Tumors One or Two Non-Coding Mutations (*n* = 46)			Group 3 - Bladder Tumors Three or More Non-Coding Mutations (*n* = 18)			*p*-Value *
	Median	Min	Max	Median	Min	Max	Median	Min	Max	Median	Min	Max	
*APOBEC3A*	1.0	0.07	13.5	0.63	0.00	17.3	0.73	0.00	7.44	0.91	0.08	18.6	**NS**
*APOBEC3B*	1.0	0.00	3.55	9.26	1.68	309	12.6	1.29	114	23.0	1.00	78.6	**0.044**
*APOBEC3H*	1.0	0.00	8.49	5.67	0.78	44.8	5.21	0.00	23.3	5.88	0.32	19.8	**NS**

* Kruskall Wallis test (no vs. 1 or 2 vs. 3 or more non-coding mutations).

**Table 5 cancers-12-02882-t005:** Relationship between TGAACA core motif mutational signature and standard clinical pathological and biological parameters.

Parameters	Total Population (%)	Group 1 No Non-Coding Mutation	Group 2 One or Two Non-Coding Mutations	Group 3 Three or More Non-Coding Mutations	*p*-Value *
Total population	103 (100.0)	39 (37.9)	46 (44.7)	18 (17.5)	
**Age (years)**					
≥60	77 (74.8)	29 (74.4)	34 (73.9)	14 (77.8)	0.95 (NS)
<60	26 (25.2)	10 (25.6)	12 (26.1)	4 (22.2)	
**Sex**					
Male	86 (83.5)	36 (92.3)	40 (87.0)	10 (55.6)	**0.0017**
Female	17 (16.5)	3 (7.7)	6 (13.0)	8 (44.4)	
**Smoking status ^a^**					
Non-smoker	28 (28.9)	6 (17.1)	16 (36.4)	6 (33.3)	0.16 (NS)
Smoker	69 (71.1)	29 (82.9)	28 (63.6)	12 (66.7)	
**History of NMIBC**					
No	59 (57.3)	27 (69.2)	20 (43.5)	12 (66.7)	**0.039**
Yes	44 (42.7)	12 (30.8)	26 (56.5)	6 (33.3)	
**Associated pTis**					
No	94 (91.3)	35 (89.7)	43 (93.5)	16 (88.9)	0.77 (NS)
Yes	9 (8.7)	4 (10.3)	3 (6.5)	2 (11.1)	
**Tumor stage**					
Cis	1 (1.0)	1 (2.6)	0 (0)	0 (0)	0.57 (NS)
Ta	30 (29.1)	11 (28.2)	17 (37.0)	2 (11.1)	
T1	15 (14.6)	5 (12.8)	6 (13.0)	4 (22.2)	
T2	19 (18.4)	8 (20.5)	8 (17.4)	3 (16.7)	
≥T3	38 (36.9)	14 (35.9)	15 (32.6)	9 (50.0)	
***FGFR3* status**					
Mutated	28 (27.2)	6 (15.4)	19 (41.3)	3 (16.7)	**0.015**
Not mutated	75 (72.8)	33 (84.6)	27 (58.7)	15 (83.3)	
***PIK3CA* status**					
Mutated	13 (12.6)	2 (5.1)	7 (15.2)	4 (22.2)	0.15 (NS)
Not mutated	90 (87.4)	37 (94.9)	39 (84.8)	14 (77.8)	
***TERT* status ^b^**					
Mutated	79 (76.7)	25 (65.8)	38 (82.6)	16 (88.9)	0.082 (NS)
Not mutated	23 (22.3)	13 (34.2)	8 (17.4)	2 (33.3)	

* Chi^2^ test; In bold: *p*-value < 0.05; ^a^ data available for 97 patients; ^b^ data available for 102 patients.

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
