# Peer review of "Genomic Instability Signature of Palindromic Non-Coding Somatic Mutations in Bladder Cancer"

_cancers, 2020, doi:10.3390/cancers12102882_

Round 1

Reviewer 1 Report

Thank you for addressing the majority of my original questions and concerns. I have just a few more somewhat minor concerns which need to be addresses;

  1. Line 118: Please provide a citation or provide more specific details regarding this method.
  2. Section 2.2.: Per convention, please provide manufacturer name, state, country for all kits, reagents etc.
  3. Section 2.7.: The authors state that is was ‘hard to assess’ RNA quantity and quality – is this due to the small quantities of RNA attained? Please clarify. Also, this statement seems to be in conflict with section 2.2. in which you state you assessed quality via running a gel. If that latter is true, then you must have had a lot of RNA. Please reconcile these statements.
  4. Line 157: Please provide a citation that support use of TBP has an endogenous control.

Author Response

Point 1: Line 118: Please provide a citation or provide more specific details regarding this method.

Response 1: As suggested by the reviewer, we added the name of the reagent used to extract the total RNA and indicated that we followed the supplier’s protocol (lines 118 and 119).

Point 2: Section 2.2.: Per convention, please provide manufacturer name, state, country for all kits, reagents etc.

Response 2: As suggested by the reviewer, manufacturer name, state and country were added for all kits and reagents used in the section 2.2. (lines 118, 121 and 123).

Point 3: Section 2.7.: The authors state that is was ‘hard to assess’ RNA quantity and quality – is this due to the small quantities of RNA attained? Please clarify. Also, this statement seems to be in conflict with section 2.2. in which you state you assessed quality via running a gel. If that latter is true, then you must have had a lot of RNA. Please reconcile these statements.

Response 3: We state that it is hard to assess RNA quantity added to each reaction mix and its quality. Indeed, we cannot certify that the efficiency of the reverse transcription reaction was identical for all samples because, on the one hand, the qualities of RNA are not strictly identical from one sample to the other, and on the other hand, the reverse transcription reactions were made in several series.

The amounts of RNA extracted were sufficient.

The quality of the RNA samples was considered sufficient for real-time quantitative RT-PCR when the 18S and 28S RNA bands were visible on agarose gel.

Point 4: Line 157: Please provide a citation that support use of TBP as an endogenous control.

Response 4: We have added one citation that support the use of TBP as an endogenous control for gene expression studies of human bladder cancer (reference 21, line 160).

Reviewer 2 Report

I checked this work and confirmed that it is of high quality and is ready for publication. 

Author Response

As suggested by the reviewers, the English in the manuscript has been thoroughly checked and edited for language and form.

English corrections are clearly highlighted using the “Track Changes” function in Microsoft Word, so that they are easily visible.

We thank you for your interest in our work and we are honoured that you have accepted to consider this Article for publication in Cancers.

This manuscript is a resubmission of an earlier submission. The following is a list of the peer review reports and author responses from that submission.

Round 1

Reviewer 1 Report

This is an interesting study, however, key background information is missing and there is insufficient discussion of the clinical relevance of their findings/potential next steps. I have listed my comments and concerns below;

  1. Abstract: This needs to be re-written to provide clarity. Please include statement following the first sentence as to what the goal of the study was, e.g. was it to identify biomarkers that can potentially be used to detect relapse/response to immunotherapy?
  2. Line 32: I assume these were patients with non-invasive BC – please clarify.
  3. Line 46: My understanding is that the use of TMB to monitor response to immunotherapy is investigational in nature, and therefore isn’t really an ‘alternative’ – this sentence should be removed or rephrased (it would be acceptable to state it was an alternative to biomarkers that are currently used in the clinic to monitor response).
  4. Introduction: additional citations need to be included for the statements made here.
  5. Introduction: please comment on the extensive literature existing for miRNA analyses/bladder cancer and provide citations.
  6. Line 74: Citation 12 doesn’t match with the statement made here – please check all of your citations to see if they correspond to the text.
  7. Line 84: please clearly state that these were patients with NMIBC
  8. Methods: Please provide specific details regarding RNA and DNA isolation as well as quantification/assessment of purity.
  9. Results: Please comment on the occurrence of mutations at these sites in normal (i.e. non-cancer) tissue
  10. Table 2 is far too small/difficult to read
  11. The authors state (lines 268-270) that ‘The outcomes of the patients in the three groups of TGAACA mutations did not differ in terms of RFS in the global population, of PFS in the NMIBC sub-group, and of RFS and OS in the MIBC sub-group’ – this does not support them as being useful clinical biomarkers, please comment on this more in the Discussion – why do the authors think this is? Was the study underpowered? Also, were correlations made with response to immunotherapy for this study? (you make reference to the fact that these biomarkers could potentially be used to predict response on lines 46/47 and so this analysis should be done/reported)
  12. Discussion, line 310: More background information needs to be provided regarding APOBEC DNA-editing proteins activity – this can be provided here or in the introduction.
  13. Discussion lines 340-41: More background information (and citations) needs to be provided regarding the use of tumor mutational burden (TMB) to monitor response to immunotherapy in bladder cancer patients – I don’t believe TMB is currently used clinically to predict response to bladder cancer (or for other cancers), please comment on this.
  14. Line 346-47: Was expression of APOBEC3B associated with worse outcome for this lung cancer study? Please provide more details re. this study/comment on the relevance of increased interferon expression.
  15. Discussion: Please comment on whether there is differential expression of the APOBEC3 genes (and GPR126 and PLEKHS1) in non-muscle invasive bladder cancer patients per the TCGA database.

Reviewer 2 Report

Manuscript entitled "Genomic instability signature of palindromic non-coding somatic mutations in bladder cancer".

This work is of interesting. However, there some major issues should be modified before final acceptance:

  1. The authors should make a summary (tables or figures) regarding the gene alterations in the cases explored by using panel NGS.
  2. The authors should disclose the association between TGAACA core motif loci alterations and gene expression pattern (signatures) of these UCs as defined by NGS. As well as the associations to the key gene alterations.